# Evaluation of the ImmuView RSV Test for Rapid Detection of Respiratory Syncytial Virus in Adult Patients with Influenza-Like Symptoms

E. Larsson,[a] S. Johansson,[b] O. Frøbert,[c,d] A. Nordenskjöld,[c] ⓘD S. Athlin[a,b]

[a]Department of Infectious Diseases, Örebro University Hospital, Örebro, Sweden
[b]School of Medical Sciences, Faculty of Medicine and Health, Örebro University, Örebro, Sweden
[c]Department of Cardiology, Faculty of Medicine and Health, Örebro University, Örebro, Sweden
[d]Department of Clinical Medicine, Aarhus University Hospital, Aarhus, Denmark

**ABSTRACT** Rapid antigen tests may enhance the diagnostic yield of respiratory syncytial virus (RSV) infections, but studies have shown low sensitivity in adults. We evaluated the novel ImmuView RSV test in adult patients with influenza-like symptoms who were prospectively enrolled at three emergency departments in two Swedish hospitals during two influenza seasons, 2017 to 2018 and 2018 to 2019. The ImmuView RSV test was performed on nasopharyngeal swabs and results were compared to those of the BinaxNOW RSV test. In the first season, tests were performed on frozen samples, while unfrozen samples were used in the second season. For comparison, tests were also performed on selected samples from children. Of 333 included adult patients, the sensitivity of ImmuView and BinaxNOW was 27% for both tests and specificities were 98% and 100%, respectively. The interassay agreement was good ($\kappa = 0.61$). There was no significant difference in test performance between frozen and unfrozen samples. In samples from children, the sensitivities of ImmuView and BinaxNOW were 67% and 70%, respectively. In conclusion, the ImmuView RSV test showed low sensitivity and high specificity for identifying RSV in adult patients with influenza-like symptoms, comparable with the BinaxNOW RSV test. Rapid RSV testing is of limited value for diagnosing RSV infection in adults.

**IMPORTANCE** By timely RSV diagnosis among patients with influenza-like symptoms, especially when influenza diagnostics turn negative, it is possible to prevent unnecessary antibiotic usage as well as reduce diagnostic testing, nosocomial transmission, and hospital stay. Previous rapid RSV tests have demonstrated poor sensitivity in adults, and we could demonstrate that the novel ImmuView RSV test similarly showed limited value for diagnosing RSV infection in adult patients. However, in contrast to many other studies, we investigated patient characteristics in cases with false-positive tests and we compared the performance between unfrozen and frozen samples. Thus, our results are important, as they generate new knowledge about rapid antigen tests.

**KEYWORDS** antigen detection, community-acquired infections, influenza, influenza-like symptoms, respiratory syncytial virus

Human respiratory syncytial virus (RSV; scientific name, *Human orthopneumovirus*) is recognized as an important pathogen causing acute respiratory tract infections (ARI) primarily in children but also in adults (1). Older adults, immunocompromised individuals, and patients with chronic cardiopulmonary disease are at risk of developing severe RSV infections (2–4). The clinical presentation varies from asymptomatic carriage to acute respiratory distress (5), similar to influenza (6). Timely RSV diagnosis may

Address correspondence to S. Athlin, simon.athlin@oru.se.

The authors declare a conflict of interest. OF: Received an unrestricted grant from Sanofi Pasteur.

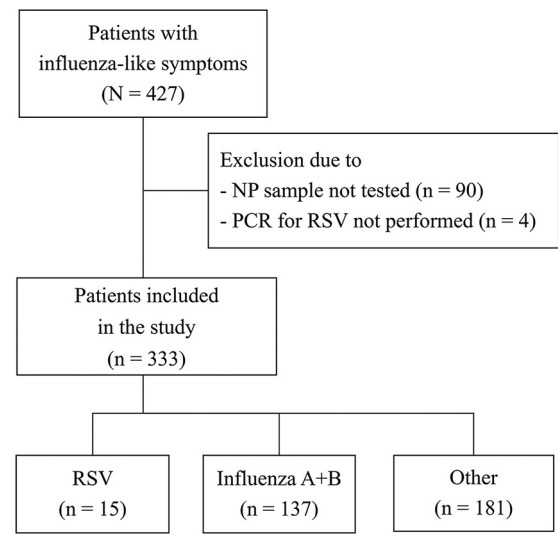

FIG 1 Flow chart of included adult patients during two influenza seasons.

prevent antibiotic usage as well as reduce diagnostic testing, nosocomial transmission, and hospital stay, especially when influenza testing is negative. Therefore, accurate and rapid RSV diagnostics are important in adult patients with influenza-like symptoms for correct management.

RSV may be identified as the causative pathogen to ARI by using sensitive molecular methods, but the routine usage of reverse transcription PCR (RT-PCR) sometimes fails to provide reliable data on disease burden alone (7). Therefore, inexpensive antigen-based methods have become important complementary methods for RSV diagnosis (8). Several assays are available for rapid RSV detection, but studies have demonstrated significantly lower sensitivity with increasing age of tested individuals (9). The aim of this study was to evaluate the performance of the novel ImmuView RSV test (SSI Diagnostica A/S, Hillerød, Denmark) on adult patients with influenza-like symptoms and compare it with that of the Alere BinaxNOW RSV Card (Abbot, Chicago, IL, USA) using RT-PCR as the reference.

## RESULTS

Of 427 eligible patients with influenza-like illness, 333 (median age, 61 years; range, 18 to 96 years; 53% female) were included in the study during the two influenza seasons (Fig. 1). Ninety-four patients were excluded because nasopharyngeal (NP) samples were unavailable ($n = 90$) or RT-PCR was not performed for RSV ($n = 4$). There was no significant difference in test performance between frozen and unfrozen samples from the two seasons, but the proportion of included patients with RSV and influenza B etiology differed (Table 1).

All tests yielded valid test results. The interassay agreement between ImmuView and BinaxNOW tests was moderate to substantial ($\kappa = 0.61$; confidence interval [CI], 0.30 to 0.92), both tests positive in four (1.2%) cases, both negative in 324 (97%) cases, and divergent results in five (1.5%) cases, all of which were positive by ImmuView and negative by BinaxNOW. The sensitivity was 27% (4/15; $P = 1.00$) for both tests and the specificities were 98% (313/318) and 100% (318/318; $P < 0.01$) for ImmuView and BinaxNOW, respectively (Table 1). Of five patients with false-positive test results by ImmuView, an etiology to infection other than RSV was identified in one case (influenza B; Table 2).

In comparison, when tests were performed on samples from 10 children (median age, <1 year; range, <1 to 10 years; 60% female) with RSV infection confirmed by RT-PCR, one ImmuView yielded an invalid test result, also after retesting. This sample

**TABLE 1** Baseline characteristics and test performance of ImmuView RSV and BinaxNOW RSV antigen tests during two influenza seasons

| Characteristic | Data for sample season/type: | | | P value |
| | All (N = 333) | 2017–2018 (frozen; n = 261) | 2018–2019 (unfrozen; n = 72) | |
|---|---|---|---|---|
| Age, median yrs (range) | 61 (18–96) | 62 (18–96) | 55 (18–94) | 0.06 |
| Female, n (%) | 178 (53) | 140 (54) | 38 (53) | 0.90 |
| Smoking, n (%) | 96 (29) | 75 (29) | 21 (29) | 0.94 |
| RSV by RT-PCR, n (%) | 15 (5) | 7 (3) | 8 (11) | 0.02 |
| Influenza A, n (%) | 65 (20) | 43 (16) | 22 (31) | 0.08 |
| Influenza B, n (%) | 72 (22) | 72 (28) | 0 (0) | <0.01 |
| Not defined etiology, n (%) | 181 (54) | 139 (53) | 42 (58) | 0.44 |
| RSV true positivity[a] | | | | |
| ImmuView, rate (%) | 4/15 (27) | 2/7 (29) | 2/8 (25) | 1.00 |
| BinaxNOW, rate (%) | 4/15 (27) | 2/7 (29) | 2/8 (25) | 1.00 |
| RSV false positivity[a] | | | | |
| ImmuView, rate (%) | 5/318 (2) | 5/254 (2) | 0/64 (0) | 0.38 |
| BinaxNOW, rate (%) | 0/318 (0) | 0/254 (0) | 0/64 (0) | 1.00 |

[a]Based on PCR for RSV as reference method.

tested positive with BinaxNOW. Based on valid test results, sensitivities of ImmuView and BinaxNOW were 67% (6/9; P = 0.09) and 70% (7/10; P = 0.03), respectively.

## DISCUSSION

In this prospective study, the ImmuView RSV test demonstrated low sensitivity (27%) but high specificity (98%) for identifying RSV in adults, and the interassay agreement with the BinaxNOW RSV test was high. Our results confirm that using these rapid tests is not a reliable method for diagnosing RSV infection in adult patients with influenza-like symptoms.

In a meta-analysis by Chartrand et al., the pooled sensitivity of rapid RSV antigen testing was 29% in adults in comparison with 81% in children, suggesting that testing should be performed in young children only (10). In comparison, when Franck et al. recently evaluated the ImmuView RSV test on retrospectively and prospectively collected samples, the sensitivity was only 12% in adults, while the sensitivity was 65% in children (11). Low sensitivity rates in adults may be explained by low viral loads (12), supported by findings of decreasing test sensitivity with increasing age (9). However, viral loads seem to be higher among adult patients with respiratory failure and,

**TABLE 2** Patient characteristics of five cases with false-positive ImmuView RSV tests on nasopharyngeal swab based on RT-PCR as the reference method[a]

| Characteristic | Patient no. | | | | |
| | 1 | 2 | 3 | 4 | 5 |
|---|---|---|---|---|---|
| Age, yrs | 42 | 62 | 67 | 76 | 85 |
| Gender | Female | Male | Female | Female | Female |
| Smoking | No | Yes | Yes | Yes | No |
| Symptom duration, days | 10 | 14 | 35 | 4 | 4 |
| Hospitalization, days | 0 | 8 | 0 | 7 | 4 |
| Antibiotic treatment | No | TMS | No | Penicillin G | No |
| Temp, °C | 39.6 | 38.7 | 39.5 | 36.7 | 39.3 |
| Chest X-ray | ND | Pos | Neg | Neg | Pos |
| C-reactive protein, mg/L | 24 | 45 | 87 | 14 | 33 |
| Leukocytes, $10^9$/L | 5.8 | 6.6 | 5.6 | 15 | 9.9 |
| Lymphocytes, $10^9$/L | 0.6 | 0.6 | 0.9 | ND | 1.7 |
| Blood culture | ND | Neg | Neg | ND | Neg |
| Airway culture | ND | Neg (NPS/S) | Neg (NPS) | ND | Neg (NPS) |
| RSV/influenza A/B RT-PCR | Neg | Neg | Neg | Neg | Influenza B |
| Influenza vaccination | No | Yes | No | Yes | No |
| ICD code | J06.9 (upper ARI) | J15.9 (bacterial pneumonia) | R50.9 (fever) | J44.1 (COPD exacerbation) | J10.1 (influenza) |

[a]ND, not done; Neg, negative; Pos, positive; RSV, respiratory syncytial virus; TMS, trimethoprim/sulfamethoxazole; ARI, acute respiratory infection; COPD, chronic obstructive pulmonary disease; ICD, International Classification of Diseases; NPS, nasopharyngeal swab; S, sputum.

accordingly, increased sensitivity of rapid RSV tests should be expected in this population (13). Using sputum for rapid testing may increase sensitivity in adults since the positive rate for respiratory virus by RT-PCR has been demonstrated to be higher than that of NP samples (14, 15).

Freezing and thawing of samples prior to testing has been suggested to reduce viral loads and affect test performance negatively (9). RSV is temperature labile and needs to be stabilized to keep its infectivity during storage in vaccine development (16), but the impact of freeze-thaw cycles on antigen detection has not been studied to the best of our knowledge. According to manufacturers, eluted swabs can be held at 2 to 8°C for up to 48 h before testing (BinaxNOW) or there are no recommendations (ImmuView), but frozen samples have been used in previous evaluation studies of both tests (11, 17). Accordingly, we observed no difference in sensitivity when testing was performed on frozen compared to unfrozen samples from two separate seasons. Similar results were reported previously when results from two different studies were compared (12). To accurately investigate the effect of sample storage on test performance, the rapid RSV tests should be evaluated on the same samples before and after controlled freeze-thaw cycles.

Of five cases with false-positive ImmuView test results on frozen samples, none was positive by BinaxNOW. One patient had a definite etiology to infection other than RSV based on extended microbiological testing performed during hospital stay. This finding is in line with previous reports of 1 to 2% false-positive test results by rapid RSV tests (9, 18, 19) and may be explained by recent RSV infection or low viral loads (20). However, repeated rapid antigen test procedures on these samples with ImmuView did not alter the results in any case, and no other method was performed to support true positivity. Thus, whether the discrepant test results by ImmuView were due to cross-reactivity or enhanced sensitivity for RSV antigen could not be determined in this study but needs to be explored further.

A limitation of the study was the low number of included adult patients positive for RSV ($n = 15$), representing only 4.5% of all included cases. In comparison, RSV was the causative pathogen in up to 12% of medically attended adults in previous studies (21). However, the rates differed between seasons (2017 to 2018, 3%; 2018 to 2019, 11%), probably due to influenza B being the dominating etiology to influenza-like symptoms in the first season, caused by a mismatch between the circulating isolate of influenza B and the annual vaccine distributed in Sweden (22). Also, the high number of included patients without RSV etiology was probably a consequence of the long inclusion period in this study, resulting in low RSV rates overall. Still, the influenza season was defined according to European surveillance systems (23) and the frequency is comparable with that found in other evaluation studies on rapid RSV testing in adults (11, 20). Another limitation is the low number of included children's samples ($n = 10$) for comparison of test performances. However, these samples were included to demonstrate that performance rates on pediatric populations were similar to those of previous studies, since children are the main target for rapid antigen testing.

In conclusion, the ImmuView RSV test showed low sensitivity and high specificity for identifying RSV in adult patients with influenza-like symptoms, comparable with the BinaxNOW RSV test. The tests performed similarly on frozen and unfrozen samples and across seasons. Due to the low detection rates, rapid RSV testing is of limited value for diagnosing RSV infection in adults.

## MATERIALS AND METHODS

**Study population.** Adult patients ≥18 years old admitted at three emergency departments at Örebro University Hospital and Karlskoga County Hospital with acute onset of influenza-like symptoms were prospectively included in this study during two influenza seasons, from October 2017 to April 2018 (first season) and from October 2018 to April 2019 (second season). NP swabs (Sigma Virocult, Medical Wire and Equipment, UK) were collected, and demographic data, clinical information, and results of routine laboratory tests at admission were recorded, including complete blood count, cultures of NP swabs, sputum, and blood, and chest radiographs within 24 h. In addition, NP swabs from children ≤10 years old were retrospectively collected at the laboratory after real-time RT-PCR had demonstrated positive results for RSV. These NP swabs had been performed at the emergency department and were used for

comparison of sensitivities of the antigen tests in this study, with no data other than age and gender available. The study was approved by the Uppsala Regional Ethical Board (dnr 2017-220, 2017-220-1, 2017-220-2, 2019-02895), and informed written consent was obtained from all individuals in accordance with the Helsinki Declaration.

**RSV detection.** NP swabs were analyzed by real-time RT-PCR (Simplexa Flu A/B & RSV, Focus Diagnostics, Cypress, CA, USA) for RSV A+B, influenza A, and influenza B within 4 h after arrival to the laboratory. During the first season, 0.5 mL of Sigma Virocult virus transport medium (VTM) from each NP swab was frozen at $-80°C$ until thawed and tested simultaneously using ImmuView and BinaxNOW in a blinded and randomized fashion. During the second season, 0.5 mL of VTM was stored at 2 to 8°C for no longer than 1 week after collection and tested simultaneously with ImmuView and BinaxNOW as described above. Briefly, both ImmuView and BinaxNOW are rapid lateral flow tests based on immuno-chromatographic technology. For ImmuView, two drops of running buffer (90 $\mu$L) were mixed with three drops of VTM (120 $\mu$L) and incubated with the test strip for 15 min at room temperature. For BinaxNOW, 100 $\mu$L of VTM was added to the test card and incubated for 15 min at room temperature. After incubation, test results were interpreted visually within 5 mins and presented as positive, negative, or invalid. Invalid test results were reanalyzed using a new test strip. This procedure was applied on samples collected from both adults and children.

**Statistical analysis.** Mann-Whitney U test was used for comparison between groups. Pearson's chi-squared test, or Fisher's exact for sample sizes lower than five, was used for comparison between proportions. McNemar's test was used for comparison of sensitivity and specificity rates performed on dependent samples. The interassay agreement between valid test results was estimated by calculating Cohen's unweighted kappa coefficient, using a CI of 95% for statistical precision. A two-tailed $P$ value of $<0.05$ was considered statistically significant. The statistical analyses were performed with the statistical software package IBM SPSS Statistics, version 27.

## ACKNOWLEDGMENTS

We thank the personnel at the department of laboratory medicine, Örebro University Hospital, for helpful assistance. The study was supported by grants from the Region Örebro County Council Research Committee (OLL-689581). ImmuView RSV and BinaxNOW RSV tests were provided by SSI Diagnostica A/S, Hillerød, Denmark.

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
