## [Reviewer comments · Microbiology Spectrum]

Microbiology Spectrum

Evaluation of the ImmuView RSV test for rapid detection of respiratory syncytial virus in adult patients with influenza-like symptoms

Emilie Larsson, Sara Johansson, Ole Frøbert, Anna Nordenskjöld, and Simon Athlin

Corresponding Author(s): Simon Athlin, Örebro University

Review Timeline:

Submission Date:	July 20, 2021
Editorial Decision:	September 1, 2021
Revision Received:	September 29, 2021
Editorial Decision:	October 6, 2021
Revision Received:	November 7, 2021
Accepted:	November 12, 2021

Editor: Rebekah Martin

Reviewer(s): Disclosure of reviewer identity is with reference to reviewer comments included in decision letter(s). The following individuals involved in review of your submission have agreed to reveal their identity: Dean D Erdman (Reviewer #2)

Transaction Report:

DOI: <https://doi.org/10.1128/Spectrum.00937-21>

September 1, 2021

Dr. Simon Athlin
Örebro University
aDepartment of Infectious Diseases, School of Medical Sciences
Södra Grev Rosengatan
Örebro 70182
Sweden

Re: Spectrum00937-21 (Evaluation of the ImmuView RSV test for rapid detection of respiratory syncytial virus in adult patients with influenza-like symptoms)

Dear Dr. Simon Athlin:

Thank you for submitting your manuscript to Microbiology Spectrum. When submitting the revised version of your paper, please provide (1) point-by-point responses to the issues raised by the reviewers as file type "Response to Reviewers," not in your cover letter, and (2) a PDF file that indicates the changes from the original submission (by highlighting or underlining the changes) as file type "Marked Up Manuscript - For Review Only". Please use this link to submit your revised manuscript - we strongly recommend that you submit your paper within the next 60 days or reach out to me. Detailed information on submitting your revised paper are below.

Link Not Available

Sincerely,

Rebekah Martin

Journals Department
Please address all reviewer comments below. In addition, please consider including the data for children in a table.

Reviewer comments:

Reviewer #1 (Comments for the Author):

In the article by Larsson et al, the authors evaluated the ImmuView RSV antigen test relative to the Alere BinaxNOW RSV antigen test, using RT-PCR testing as the reference method. Overall the authors data demonstrates poor sensitivity for the ImmuView RSV and BinaxNOW RSV antigen tests in adults. Both assays performed better (sensitivity of 66 and 70%) in pediatric patients but still not equivalent to RT-PCR. Finally, there were 5 false-positive ImmuView RSV results, resulting in a specificity of 98% versus the 100% specificity for the BinaxNOW RSV.

A. On line 44, it states that "influenza diagnostics turn negative". It would be better to say "influenza testing was negative" or something similar.

B. On line 68, it states patients got "cultures of NP swabs". What pathogens were cultured for from NP swabs?

C. In the materials and methods section, please include a description of how the BinaxNOW RSV testing was performed, as was done with the ImmuView RSV. Per the package inserts for these tests, is freezing and then thawing or refrigerating specimens up 7 days before testing acceptable? Please add this information to the Discussion section.

D. On line 125, the BinaxNOW sensitivity should be 66% (6/9) not 60%.

E. On lines 160-162, it is noted that "repeated test procedures on these samples did not alter the results" in regards to the false positive RSV results from the ImmuView RSV test. Did the authors repeat the antigen testing for these false-positive specimens for both assays and they RT-PCR? Please clarify.

Reviewer #2 (Comments for the Author):

Manuscript ID: Spectrum 00937-21

Manuscript title: Evaluation of the ImmuView RSV test for rapid detection of respiratory syncytial virus in adult patients with influenza-like symptoms

Summary and reviewer general comment:

The manuscript is well written and the study design reasonable (the number of RSV PCR positive adults tested is low, but this is as acknowledged by the authors). The findings are not new, but confirmatory, as antigen detection insensitivity in adults is well recognized for most respiratory viruses.

Specific reviewer comments:

- 1) Introduction, Line 49. Change PCR to RT-PCR here and throughout.
- 2) Methods, Line 80, 82. Does "liquid medium" mean commercial viral transport media?
- 3) Results, Line 113. I would consider changing the term "good" to "moderate to substantial" agreement for a kappa of 0.61.
- 4) Table 2 is not essential and could be removed. It should be noted that the possibility that another viral/bacterial etiology might be responsible for the 5 false positive ImmuView test results, but could not be determined here as only influenza A/B was tested for by RT-PCR.
- 5) A more comprehensive assessment of the false negative test results of both antigen assays would have examined the raw RT-PCR data (Ct values) to see if low virus loads account for antigen detection failures.

Staff Comments:

Preparing Revision Guidelines

Please return the manuscript within 60 days; if you cannot complete the modification within this time period, please contact me. If you do not wish to modify the manuscript and prefer to submit it to another journal, please notify me of your decision immediately so that the manuscript may be formally withdrawn from consideration by Microbiology Spectrum.

If you would like to submit an image for consideration as the Featured Image for an issue, please contact Spectrum staff.

Response to Reviewers

Please address all reviewer comments below. In addition, please consider including the data for children in a table.

Response: Thank you for this suggestion. However, we consider it is better not to include children's data in existing tables of adults since the main comparison is between adults during two seasons and the number of children is small. Also, existing data only require a small space as written in text and would need to be added as footnotes in a table. Therefore, after we have tried to merge the data in tables in different ways, we would prefer to keep it as it is if possible.

Reviewer #1 (Comments for the Author):

In the article by Larsson et al, the authors evaluated the ImmuView RSV antigen test relative to the Alere BinaxNOW RSV antigen test, using RT-PCR testing as the reference method. Overall the authors data demonstrates poor sensitivity for the ImmuView RSV and BinaxNOW RSV antigen tests in adults. Both assays performed better (sensitivity of 66 and 70%) in pediatric patients but still not equivalent to RT-PCR. Finally, there were 5 false-positive ImmuView RSV results, resulting in a specificity of 98% versus the 100% specificity for the BinaxNOW RSV.

A. On line 44, it states that "influenza diagnostics turn negative". It would be better to say "influenza testing was negative" or something similar.

Response: The text was changed.

B. On line 68, it states patients got "cultures of NP swabs". What pathogens were cultured for from NP swabs?

Response: Cultures of NP swabs, sputum and blood were performed in clinical routine. For NP swabs, identification of common airway pathogens causing lower and upper airway infections was performed, such as *Streptococcus pneumoniae*, *Haemophilus influenzae*, *Moraxella catarrhalis* and hemolytic streptococci. Other species, such as *Staphylococcus aureus*, *Pseudomonas aeruginosa*, *Neisseria meningitidis* etc are reported depending on quantity and reported medical conditions, such as immunosuppression, cystic fibrosis.

C. In the materials and methods section, please include a description of how the BinaxNOW RSV testing was performed, as was done with the ImmuView RSV. Per the package inserts for these tests, is freezing and then thawing or refrigerating specimens up 7 days before testing acceptable? Please add this information to the Discussion section.

Response: This was added to the material and method section and to the discussion section, respectively.

D. On line 125, the BinaxNOW sensitivity should be 66% (6/9) not 60%.

Response: The sensitivity was changed.

E. On lines 160-162, it is noted that "repeated test procedures on these samples did not alter the results" in regards to the false positive RSV results from the ImmuView RSV test. Did the authors

repeat the antigen testing for these false-positive specimens for both assays and they RT-PCR? Please clarify.

Response: We repeated the rapid antigen test procedure for ImmuView only. This was clarified in the text.

Reviewer #2 (Comments for the Author):

Manuscript ID: Spectrum 00937-21

Manuscript title: Evaluation of the ImmuView RSV test for rapid detection of respiratory syncytial virus in adult patients with influenza-like symptoms

Summary and reviewer general comment:

The manuscript is well written and the study design reasonable (the number of RSV PCR positive adults tested is low, but this is as acknowledged by the authors). The findings are not new, but confirmatory, as antigen detection insensitivity in adults is well recognized for most respiratory viruses.

Specific reviewer comments:

1) Introduction, Line 49. Change PCR to RT-PCR here and throughout.

Response: This was corrected throughout the manuscript.

2) Methods, Line 80, 82. Does "liquid medium" mean commercial viral transport media?

Response: Yes, this was clarified in the text.

3) Results, Line 113. I would consider changing the term "good" to "moderate to substantial" agreement for a kappa of 0.61.

Response: This was changed according to suggested definition.

4) Table 2 is not essential and could be removed. It should be noted that the possibility that another viral/bacterial etiology might be responsible for the 5 false positive ImmuView test results, but could not be determined here as only influenza A/B was tested for by RT-PCR.

Response: Thank you for the recommendation. We are aware of that extended methodology needs to be performed in order to cover for the most common etiologies to influenza-like illness, including wide panels for viral and bacterial identification. However, data on false-positive test results are rarely presented to let the reader objectively assess the clinical picture of patients tested. Therefore, we wish to present these data to extend the information on cases responsible for false-positive test results by ImmuView.

5) A more comprehensive assessment of the false negative test results of both antigen assays would have examined the raw RT-PCR data (Ct values) to see if low virus loads account for antigen detection failures.

Response: Thank you for this suggestion, we agree that Ct values may have strengthened the discussion regarding false-positive and false-negative results. However, Ct values were not available in this study.

October 6, 2021

Dr. Simon Athlin
Örebro University
aDepartment of Infectious Diseases, School of Medical Sciences
Södra Grev Rosengatan
Örebro 70182
Sweden

Re: Spectrum00937-21R1 (Evaluation of the ImmuView RSV test for rapid detection of respiratory syncytial virus in adult patients with influenza-like symptoms)

Dear Dr. Simon Athlin:

Thank you for submitting your manuscript to Microbiology Spectrum. I have included comments below that require further clarification for this manuscript. Please submit a revised version addressing these questions. When submitting the revised version of your paper, please provide (1) point-by-point responses to the issues raised by the reviewers as file type "Response to Reviewers," not in your cover letter, and (2) a PDF file that indicates the changes from the original submission (by highlighting or underlining the changes) as file type "Marked Up Manuscript - For Review Only". Please use this link to submit your revised manuscript - we strongly recommend that you submit your paper within the next 60 days or reach out to me. Detailed information on submitting your revised paper are below.

Link Not Available

Sincerely,

Rebekah Martin

Journals Department
Editorial comments:

All lines indicated are from the merged document.

Line 29: I believe this also needs to be changed from 60% to 67%.

Line 44: "was negative" should be changed to "is negative". Apologies for not seeing this on the first draft.

Line 69: "from clinical routine" is unclear. Please clarify.

Line 80: Following up on original reviewer comments (reviewer 2, comment #2)--- there are a number of liquid transport media available. What was acceptable for samples to arrive in? Saline? UTM? VTM? Something else? Please specify which type(s) of liquid transport media.

Line 166-7: As a follow up to original reviewer comments (reviewer 2, comment #4) --- authors mention "no other method

supported true positivity". Any additional methods used here should be included in the methods section. Was RT-PCR repeated? If so, that should be stated. If not, what are the "other methods" referenced here?

Reviewer 2, comment #5: regarding false negative test results and CT values. Would authors be able to comment on the limit of detection (LOD) for the antigen assays vs the RT PCR assay? Although CT values are not available, having an understanding of the LODs for each assay may help make a similar point if the RT PCR assay has a much lower LOD.

Staff Comments:

Preparing Revision Guidelines

Please return the manuscript within 60 days; if you cannot complete the modification within this time period, please contact me. If you do not wish to modify the manuscript and prefer to submit it to another journal, please notify me of your decision immediately so that the manuscript may be formally withdrawn from consideration by Microbiology Spectrum.

Response to Reviewers

Editorial comments:

All lines indicated are from the merged document.

Line 29: I believe this also needs to be changed from 60% to 67%.

Response: This was corrected.

Line 44: "was negative" should be changed to "is negative". Apologies for not seeing this on the first draft.

Response: This was changed.

Line 69: "from clinical routine" is unclear. Please clarify.

Response: The NP samples from adults were prospectively collected according to the study protocol for RT-PCR and antigen testing, while the NP samples from children were retrospectively collected at the laboratory after real-time RT-PCR had demonstrated positive results for RSV. These NP swabs had been performed at the emergency department and were used for antigen testing for comparison of sensitivities of the antigen tests in this study. This was clarified in the text.

Line 80: Following up on original reviewer comments (reviewer 2, comment #2)--- there are a number of liquid transport media available. What was acceptable for samples to arrive in? Saline? UTM? VTM? Something else? Please specify which type(s) of liquid transport media.

Response: Thank you for following up this remark, as we had a chance to add this information, but also as we became aware of that the swab manufacturer was incorrectly stated in the methods section. We used Sigma Virocult Virus Transport Medium during the study period, containing 1.0 mL balanced salt medium, buffered with disodium hydrogen orthophosphate, which also contains lactalbumin hydrolysate as a stabilizer, and chloramphenicol and amphotericin to inhibit the growth of any bacterial contaminants in the specimen. This was corrected.

Line 166-7: As a follow up to original reviewer comments (reviewer 2, comment #4) --- authors mention "no other method supported true positivity". Any additional methods used here should be included in the methods section. Was RT-PCR repeated? If so, that should be stated. If not, what are the "other methods" referenced here?

Response: Thank you for this remark, this formulation was unfortunately overlooked in the review of the manuscript. It should say that no other method was performed to support true positivity. This was clarified in the text.

Reviewer 2, comment #5: regarding false negative test results and CT values. Would authors be able to comment on the limit of detection (LOD) for the antigen assays vs the RT PCR assay? Although CT values are not available, having an understanding of the LODs for each assay may help make a similar point if the RT PCR assay has a much lower LOD.

Response: Thank you for letting us comment on this matter. It is difficult to compare the LOD between RT-PCR and the two antigen tests, since equal data for each test is not available according to our knowledge, and the RSV strain in each case is not identified in this study. According to each package insert, the Simplexa RT-PCR has a LOD at 1-3 TCID₅₀/mL for RSV, depending on RSV strain. The ImmuView® RSV antigen test has a LOD at 1.77 µg/mL. and for inactivated native RSV strain A it is 1.25x10⁵ TCID₅₀/mL. The BinaxNOW antigen test has a LOD at 1.56 x 10⁻¹ to 5.00 x 10⁴ TCID₅₀/ml for 6 RSV A and 5 RSV B strains. If possible, we prefer not to add all this information, since it will not add understanding to any discrepancies among LODs, as long as the RSV strains in each patient in this study is not identified.

November 12, 2021

Dr. Simon Athlin
Örebro University
aDepartment of Infectious Diseases, School of Medical Sciences
Södra Grev Rosengatan
Örebro 70182
Sweden

Re: Spectrum00937-21R2 (Evaluation of the ImmuView RSV test for rapid detection of respiratory syncytial virus in adult patients with influenza-like symptoms)

Dear Dr. Simon Athlin:

Your manuscript has been accepted, and I am forwarding it to the ASM Journals Department for publication. You will be notified when your proofs are ready to be viewed.

Sincerely,

Rebekah Martin
Editor, Microbiology Spectrum
